# Nuclear Lipid Droplet Birth during Replicative Stress

**DOI:** 10.3390/cells11091390

**Published:** 2022-04-20

**Authors:** Sylvain Kumanski, Romain Forey, Chantal Cazevieille, María Moriel-Carretero

**Affiliations:** 1Centre de Recherche en Biologie Cellulaire de Montpellier (CRBM), Université de Montpellier-Centre National de la Recherche Scientifique, 34293 Montpellier, France; sylvain.kumanski@crbm.cnrs.fr; 2School of Life Sciences, École Polytechnique Fédérale de Lausanne (EPFL), 1015 Lausanne, Switzerland; romain.forey@epfl.ch; 3Institut de Neurosciences de Montpellier (INM), Université de Montpellier-INSERM, 34091 Montpellier, France; chantal.cazevieille@inserm.fr

**Keywords:** nuclear lipid droplets, replicative stress, unsaturated fatty acids, sterols

## Abstract

The nuclear membrane defines the boundaries that confine, protect and shape the genome. As such, its blebbing, ruptures and deformations are known to compromise the integrity of genetic material. Yet, drastic transitions of the nuclear membrane such as its invagination towards the nucleoplasm or its capacity to emit nuclear lipid droplets (nLD) have not been evaluated with respect to their impact on genome dynamics. To begin assessing this, in this work we used *Saccharomyces cerevisiae* as a model to ask whether a selection of genotoxins can trigger the formation of nLD. We report that nLD formation is not a general feature of all genotoxins, but of those engendering replication stress. Exacerbation of endogenous replication stress by genetic tools also elicited nLD formation. When exploring the lipid features of the nuclear membrane at the base of this emission, we revealed a link with the unsaturation profile of its phospholipids and, for the first time, of its sterol content. We propose that stressed replication forks may stimulate nLD birth by anchoring to the inner nuclear membrane, provided that the lipid context is adequate. Further, we point to a transcriptional feed-back process that counteracts the membrane’s proneness to emit nLD. With nLD representing platforms onto which genome-modifying reactions can occur, our findings highlight them as important players in the response to replication stress.

## 1. Introduction

The nucleoplasmic space is crowded by the genetic material and the factors involved in its exploitation and integrity. The volume allocated to this compartment is defined by the nuclear envelope, and finely regulated by the gating and transfer of molecules in and out of this confinement by the nuclear pores. The nuclear membrane can suffer ruptures, blebbing, deformations and invaginations [1], as well as emit nuclear lipid droplets (nLD) that occupy the intra-nuclear space [2], all of which will alter the volume (and consequently the territorial organization) of the nucleus and its genetic material. Cytoplasmic LD can form between the phospholipid tails of the endoplasmic reticulum and/or the outer nuclear membrane leaflets through the concentration of neutral fats and sterols that, when reaching a threshold concentration, induce their budding towards the cytoplasm [3]. In contrast, the process through which LD form and bud towards the nucleus does not necessarily follow this sequence: first, the composition of the inner nuclear membrane (INM) phospholipids is less well-known. Second, the LD is born from a positive curvature when forming towards the cytoplasm, while it forms from a negatively curved surface when emerging towards the nucleoplasm. Third, not all the enzymes known to catalyze neutral lipids esterification are necessarily present at the INM. For example, while in *S. cerevisiae* nLD formation from the INM can be reminiscent to that of their cLD counterparts [4], cells of hepatic origin form nLD from ApoB precursors accumulated in the lumen of the endoplasmic reticulum that manage to access the nucleus coated by a membrane derived from the INM [5].

Among the most studied phenomena that can alter the genetic material is the use of genotoxins, which directly or indirectly harm DNA integrity, thus putting at risk the stability of the genome. Their use has granted an expansion of knowledge on how DNA can be harmed and repaired, the consequences for the cell if this repair does not take place and the rescue mechanisms that exist because of an intricate crosstalk. This knowledge is important for treatment design. Particularly relevant genotoxins for chemotherapy purposes are those generating replication stress [6]. This concept gathers, in a broad manner, any interference with the process of genome duplication, which includes physical blocks ahead of the advancing replication fork, such as protein–DNA adducts, mutations affecting polymerase processivity, shortages in nucleotides or conflicts when sharing the DNA template with other machineries, such as transcription polymerases [7].

Despite our understanding of how genotoxins directly harm DNA or trigger replication stress, less is known about how they might indirectly affect other aspects of nuclear biology, which may yet prove relevant to fully account for their mode of action. In this work, we have used *Saccharomyces cerevisiae* as a model to ask whether a selection of genotoxins elicit the formation of nLD. We report that nLD formation is not a general feature of all genotoxins, but is a feature of those that engender replication stress. Exacerbation of endogenous replication stress by genetic tools also elicited nLD formation. We additionally explored the lipid features of the nuclear membrane at the base of this emission and revealed a link with the unsaturation profile of its phospholipids and, for the first time, of its sterol content. We propose the notion that physical “pinching” of the INM by stressed replication forks can prime the emission of nLD, and also suggest a role for free sterols within membranes in favoring this transition.

## 2. Materials and Methods

Cell culture: *Saccharomyces cerevisiae* cells were grown at 25 °C in YEP (rich) or yeast nitrogen base (YNB) (minimal) liquid medium supplemented with 2% glucose (dextrose). Transformed cells selected for biosensor plasmid maintenance were grown in YNB–leucine overnight. In this case, in the morning, the exponentially growing cultures were mildly diluted and grown for at least 4 additional hours in rich medium prior to exposure to the genotoxic agents with the goal of optimizing drug entry into the cells. To synchronize strains, 2.15 µg/mL α-factor was added to the exponentially growing cultures; 1 h later a further 4.3 µg/mL was added; then, after an additional 1 h, 2.15 µg/mL was added again for the last 45 min. Synchronization success was verified by microscopy prior to release. Block release from G_1_ was achieved by cell filtration and washing, plus the addition of 40 mM potassium phosphate buffer (pH 7) and 75 µg/mL pronase to the fresh culture medium. The strains and the plasmids used in this study are detailed in Table 1 and Table 2, respectively. Plasmid pQ2 was obtained by removing the nuclear localization signal (NLS) from pNLS-Q2 by digestion with *Xba*I and subsequent religation. Tagging of Pus1 with mCherry was achieved by digestion of pmCherry-*PUS1* with *Bgl*II and cell transformation of the linearized fragment followed by selection in YNB-ura plates.

Reagents: 4-nitroquinoline-1-oxide (4-NQO, N8141, Sigma-Aldrich, Saint-Quintin-Fallavier, France), methylmetane sulfonate (MMS, 129925, Sigma-Aldrich, Saint-Quintin-Fallavier, France), zeocin (R25001, Thermo Fisher Scientific, Illkirch-Graffenstanden, France), hydroxyurea (HU, H8627, Sigma-Aldrich, Saint-Quintin-Fallavier, France), camptothecin (CPT, C9911, Sigma-Aldrich, Saint-Quintin-Fallavier, France), BODIPY (790389, Sigma-Aldrich, Saint-Quintin-Fallavier, France), NileRed (HY-D0718, CliniSciences, Nanterre, France), pronase (53702-25KU, Sigma-Aldrich, Saint-Quintin-Fallavier, France).

Fluorescence microscopy: 1 mL of the culture of interest was centrifuged; the supernatant was thrown away and the pellet was resuspended in the remaining 50 µL. Next, 3 µL of this cell suspension was directly mounted on a coverslip for immediate imaging of the pertinent fluorophore-tagged protein signals. To dye LD, 1 µL of a 100 µg/mL BODIPY stock or 1 µL of a 1 mg/mL NileRed stock was added to the centrifuged pellet and the remaining 50 µL supernatant prior to imaging. Imaging was conducted using a Zeiss Axio Imager Z2 microscope (Carl Zeiss S.A.S., Rueil Malmaison, France) and visualization, co-localization and visual inspection by the experimenter were performed with Image J (v2.0.0-rc-69/1.52i). To perform through-focus series analyses, cells were fixed with 4% PFA at room temperature for 20 min, and through-focus series acquired every 0.26 µm for 8 µm. Deconvolution was conducted using Huygens professional v21.04 (Scientific Volume Imaging, Hilversum, the Netherlands) using the Cmle algorythm. Three-dimensional view reconstructions and slices were obtained using Imaris v9.8 (Oxford Instruments, Abingdon, United Kingdom). Determination of the percentage of cells in the population displaying nLD was performed by visual counting by the experimenter.

Electron microscopy: Cells were immersed in a solution of 2.5% glutaraldehyde in 1× PHEM buffer (pH 7.4) overnight at 4 °C. They were then rinsed in PHEM buffer and post-fixed in 0.5% osmic acid + 0.8% potassium hexacyanoferrate trihydrate for 2 h in the dark at room temperature. After two rinses in PHEM buffer, the cells were dehydrated in a graded series of ethanol solutions (30 to 100%). Cells were embedded in EmBed 812 using an Automated Microwave Tissue Processor for Electronic Microscopy, Leica EM AMW (Leica Microsystems S.A.S., Nanterre, France). Thin sections (70 nm) were collected at different levels of each block. These sections were counterstained with 1.5% uranyl acetate in 70% ethanol and lead citrate and observed using a Tecnai F20 transmission electron microscope (Thermo Fisher Scientific, Villebon-sur-Yvette, France) at 120 KV located in the Institut des Neurosciences de Montpellier, Electronic Microscopy facilities, INSERM U1298, Université Montpellier, Montpellier, France.

Cytometry: 430 µL of each culture sample at 10^7^ cells/mL was fixed with 1 mL of 100% ethanol. Cells were centrifuged for 1 min at 16,000× *g* and resuspended in 500 µL 50 mM Na-Citrate buffer containing 5 µL of RNase A (10 mg/mL, Euromedex, RB0474, Strasbourg, France) for 2 h at 50 °C. Then, 6 µL of Proteinase K (Euromedex, EU0090-C, Strasbourg, France) was added for 1 h at 50 °C. Aggregates of cells were dissociated by sonication (one 3 s pulse at 50% potency in a Vibracell 72405 Sonicator). After this, 20 µL of this cell suspension was incubated with 200 µL of 50 mM Na-Citrate buffer containing 4 µg/mL Propidium Iodide (P4170, Thermo Fisher Scientific, Illkirch-Graffenstanden, France). Data were acquired and analyzed on a Novocyte Express (Novocyte, Agilent, Les Ulis, France).

Bioinformatics: The dataset GSE6018 was re-analyzed according to [8], with modifications. In particular, the “MM” signal was subtracted from the corresponding “PM” signal for each probe pair, where perfect match probe (PM) indicates perfect matching 25 mer oligos to the target transcripts, and mismatch probes (MM) contain sequences with the 13th position of the corresponding PM sequence that are modified to the complement nucleotide as to reduce the effect of non-specific hybridization signals when estimating transcript abundance. The data from the 15 probe pairs specific for each open reading frame were averaged and genes with a low or no-negative signal were filtered out. This provided us with the raw intensity table (Appendix A). The raw intensity table has been used to compare MMS-treated samples to untreated samples, computing differential expression analysis using Limma [9] (script available as Appendix A). This provided us with differential analysis tables for each condition compared to the control (Appendix A), which we merged and used to build a heatmap (script available as Appendix A). Using this output, we defined as differentially expressed the genes with an adjusted *p* value < 0.01. We used differentially expressed genes to perform Gene Ontology (GO) using PANTHER software version 16 [10] (Appendix A).

Graphical representation and Statistical Analysis were conducted using GraphPad Prism (GraphPad Software, Inc., v9.3.1, San Diego, CA, USA), R studio and Microsoft Excel version 16.16.27.

We regularly present independent biological replicates by a dot of a given color. This way, if cells in a culture were monitored over time after a treatment (or the lack of it), the reader can follow the result over time for each independent experiment. Further, for mean population values obtained out of a cloud of points (for example the number of lipid droplets per cell, or per nucleus), we also include in the same graph all the individual measured values for each independent experiment in a lighter shade of the same color. This gives rise to at least three overlapping clouds of dots and their respective means, a SuperPlot [11], which fully illustrates the reproducibility (or lack of it) in an experiment and its replicates. Of note, for a fair evaluation of the significance of differences, we perform statistical analyses using as independent values the three to six means from each independent experiment, and not the 300 values that belong to one single experiment [11].

**Table 2 cells-11-01390-t002:** Plasmids used in this study.

Simplified Name	Detailed Information	Source
pNLS-Q2	pRS316-*SacI-SacII*-*CYC1*promoter (truncated)-*XbaI*-*NUP60NLS*(1-24)-*XbaI*-Opi1 Q2 -*BamHI*-mCherry-*XmaI*-*NUP1* terminator-*XhoI*	[4]
pQ2	pRS316-*SacI-SacII*-*CYC1*promoter (truncated)- *XbaI*-Opi1 Q2 -*BamHI*-mCherry-*XmaI*-*NUP1* terminator-*XhoI*	This study
pEmpty	pRS424	[12]
p*exo1-D173A*^OE^	pSM638 (*pRS424-exo1-D173A*)	[13]
p*EXO1*^OE^	pSM502 (*pRS424-EXO1*)	[13]
pmCherry-*PUS1*	YIplac211-mCherry-*PUS1*	Symeon Siniossoglou

## 3. Results

### 3.1. The Formation of Nuclear Lipid Droplets Is Not a General Response to Genotoxins

To ask whether the nucleoplasmic space can be occupied by nLD during treatment with genotoxins, we used a WT strain in which the nucleoplasm could be observed by means of Pus1 tagging with mCherry, while LD were observed by staining them with BODIPY (Figure 1A). We made sure that the considered events were not cytoplasmic LD in close proximity to the nucleus, which might have impacted our quantification. To this end, we acquired through-focus serial images of cells, deconvolved them using Huygens, and performed 3D reconstructions with Imaris. Further, we referred to published work to make sure that the observed size for the structures of interest matched that reported for nLD (approx. 100 to 400 nm, Ref. [4]). This analysis clearly showed that the events we visually designated as nLD were fully inside the nucleus (Figure 1B and Appendix A).

We then systematically assessed the percentage of cells bearing at least one nLD both in the untreated asynchronous population and in cells exposed for 120 and 210 min to a selection of genotoxins, each known to harm DNA differently. This included camptothecin (CPT), which traps the topoisomerase I on DNA after cutting one strand; the alkylating agent methylmetane sulfonate (MMS); the nucleotide-harming agent 4-nitroquinoline n-oxide, 4-NQO; the inhibitor of the synthesis of deoxynucleotides hydroxyurea, HU; and the radiomimetic agent zeocin. The effect of the treatments was apparent from the halt in cell cycle progression that we could observe upon cytometry analysis (Figure 1C, red asterisks). Although nLD are described as rare or absent in the WT strain [4,14], we observed a basal level of 10% of cells bearing at least one of them (Figure 1D). We observed a very modest and non-significant effect in the percentage of cells displaying nLD when cells were exposed to 4-NQO or zeocin (Figure 1D). These treatments had the property of arresting cell cycle progression at the G_2_ phase (Figure 1C). The addition of CPT, which creates damage at progressing forks during replication that are nevertheless processed later during G_2_ [15], led to a mild yet significant increase in response to CPT (Figure 1C). As for the addition of HU or MMS, we observed a marked time-dependent accumulation of nLD in the cells (Figure 1D). Although they harm DNA through different mechanisms, HU and MMS both perturb the progression of replication forks and, as such, lead to the accumulation of cells within the S-phase of the cell cycle (Figure 1C). Of note, within those nuclei containing at least one LD, the mean number of LD (which was slightly above one in the untreated condition) did not significantly increase in response to any of the treatments (Appendix A). Together, these observations suggest the following: (a) nLD induction is not a feature of DNA damage per se, but the birth of nLD is somehow more prone to occur during the replication phase of the cell cycle and/or during replication stress; (b) whatever underlies nLD formation in such situation(s) is spatially constrained.

### 3.2. nLD Form in Response to Replication Stress

A DNA damage-independent yet replication fork-related nLD emergence points to a physical interaction between the DNA molecule challenged during replication and the membrane from which the nLD are born. On the one hand, troubled replication forks have previously been associated with the nuclear membrane [16,17,18,19]. On the other hand, the physical tethering exerted by nucleic acid-protein complexes contacting the membrane can generate a force that pulls membrane-derived structures towards the nucleoplasm [5,20,21]. We therefore postulated that stalled replication forks anchored to the nuclear membrane could physically stimulate nLD birth. Accordingly, electron microscopy revealed that the nLD in cells that had been treated with MMS for 210 min were in tight contact with the INM, which in some cases even seemed stretched away from the outer nuclear membrane (Figure 1E, example 2, yellow arrowheads). This mode of birth is in full agreement with previous descriptions [4].

To further explore the possibility that stalled replication forks may “pinch” the nuclear membrane, thus stimulating nLD emission, we turned to replication stress that was induced genetically. We used mutants lacking either the nuclease Exo1, the helicase Sgs1, or both of these, since the absence of these proteins, needed to stabilize stalled or damaged forks, elicits replication stress due to endogenous problems [22,23]. We observed that, basally, *sgs1∆*, *exo1∆* and *sgs1∆exo1∆* cells displayed an elevated level of nLD that was (1) identical between these cells; (2) stable over time; and (3) equal to that induced by HU or MMS during 210 min in the WT strain (Figure 2A). Of note, G_1_-synchonization and subsequent release experiments showed that single mutants progress through the S-phase more synchronously than WT cells, and that all three strains can complete their S-phase transition within the same time, namely 45 min (Figure 2B). This suggests that nLD may be related to replicative stress but this is not merely a result of spending longer in the S-phase. 

We then combined HU addition with *sgs1∆*, *exo1∆* or *sgs1∆exo1∆* cells. This did not increase the percentage of nLD-bearing cells (Figure 2C, left). Treatment with MMS led to a further significant increase of the phenotype, both in the single and in the double mutants (Figure 2C, right). We thus note that, in spite of high loads of replicative stress, such as that suffered by the double mutant exposed to MMS, it appears difficult to observe nLD birth in more than 50% of the population. To explore whether a functional link underlined replication stress and nLD formation, we overexpressed Exo1 in cells in which replicative stress was elicited either by exposure to HU or to MMS. Both drugs triggered the formation of nLD in the strain harboring the control empty plasmid to an extent comparable to that induced in the untransformed cells (Figure 2D, pEmpty, compare to Figure 1D, untreated). Importantly, overexpression of Exo1 greatly lessened this increase (Figure 2D, p*EXO1*^OE^), while the overexpression of the catalytically dead nuclease *Exo1-D173A* [13,24] did not prevent nLD formation in response to the same genotoxins (Figure 2D, p*exo1-D173A*^OE^). Thus, nLD formation seems to be elicited by replicative stress.

### 3.3. Replication Stress-Associated nLD Birth Is Not Accompanied by Biosensor-Detectable Phosphatidic Acid Changes at the Nuclear Membrane

The observation that the phenomenon of nLD formation is unlikely to occur in more than 50% of the population (Figure 2C) made us reason that the pulling force exerted by stressed forks is perhaps insufficient to induce nLD budding per se, or at least that it could be further influenced by specific membrane features. LD are normally born from the cytoplasmic leaflet of the outer nuclear membrane towards the cytoplasm in *S. cerevisiae*, and this directionality is naturally favored by the cell over the scenario where LD are born from the nucleoplasmic leaflet of INM towards the nucleoplasm [25]. At present, two main factors are defined as responsible for nLD formation in *S. cerevisiae*: either an increase in the presence of phosphatidic acid (PA) in the nuclear membrane, upon which more LD form in both directions, namely towards the cytoplasm and the nucleoplasm; or an increase in the degree of unsaturation of the inner nuclear membrane phospholipids that gives rise to, exclusively, more nLD [25]. We first explored the potential enrichment of PA at the nuclear membrane by using two versions of a fluorescent PA biosensor (description in the legend of Figure 3A). If expressed unmodified, this sensor mainly binds the plasma membrane and yields a faint nucleoplasmic signal, as described [26], whereas, if fused to a NLS, it provides exclusively a nucleoplasmic diffuse signal (Figure 3A, [4]). Their specificity and performance can be validated by genetically forcing the accumulation of PA (for example, in the absence of the PA phosphatase Pah1), which, apart from a very strong deformation of the nuclear contour, causes the binding of the sensor to the nuclear envelope (Figure 3A, arrowheads). We treated exponentially growing cells with MMS and monitored the localization of both PA biosensor signals over time. We failed to detect the accumulation of the biosensors at the nuclear envelope at any of the assayed times, which included 30, 60, 100, 150 and 210 min (Figure 3B). This suggests that PA enrichment at the nuclear envelope may not underlie the formation of LD towards the nucleoplasm in response to MMS. In agreement, while PA increase at the nuclear membrane elicits LD formation in both directions, namely towards the cytoplasm and the nucleoplasm [4,25], we observed that nLD birth in response to MMS was not accompanied by any global increase of LD in the whole cell, since the number of LD per cell remained mostly flat over time (Figure 3C). Thus, PA does not appear to relate to nLD formation during MMS treatment.

### 3.4. The Membrane Phospholipid Unsaturation and Sterol Profile Conditions nLD Birth in Response to Replicative Stress

Our data therefore suggested that the increase of LD in the nucleus in response to replicative stress in general, and MMS in particular, may relate to a preferential increase in nLD exclusively from the INM. This is described as the consequence of an increase in the unsaturation degree of the nuclear membrane phospholipids, itself linked to the transcriptional upregulation of the desaturase Ole1 and of the sterol biosynthesis enzyme Mvd1 [25]. Of them, Ole1 is the only cellular enzyme capable of desaturating fatty acids and, given that there exists no opposing enzimatic activity, Ole1 half-life control is the only means of regulating unsaturation profiles in the cell [27]. We exploited publicly available transcriptomes of *S. cerevisiae* cells exposed for 1 h to increasing doses of MMS (0.001%, 0.01%, 0.1%) for which three biological replicates have been performed with high reproducibility ([8] and Appendix A). In agreement with [8], exposure to MMS leads to transcriptomic alterations in a dose-dependent manner with an equivalent number of both downregulated and upregulated genes (Appendix A). Genes dysregulated upon 0.001% MMS were also dysregulated at 0.01% and 0.1% MMS, and were therefore considered highly responsive to MMS (Appendix A). To document the pathways dysregulated in the presence of MMS, we ran a Gene Ontology (GO) analysis of downregulated or upregulated genes separately, and for each of the three doses of MMS (Figure 4A and Appendix A).

In this analysis, we made the striking observation that lipid metabolic pathways involved in sterol metabolism were among the top downregulated pathways in response to the three doses of MMS (Figure 4B, Appendix A). Among the genes showing a dose-dependent downregulation, we found that most are involved in the ergosterol metabolic pathway (Figure 4C), in particular related to sterol synthesis (Table 3, in bold), including *MVD1* (Figure 4D). We also retrieved *OLE1* as our second-best hit (Figure 4D; Table 3, in bold). Thus, these data support the possibility that a transcriptional program is launched early in response to MMS (and potentially to replicative stress) that counteracts membrane unsaturation and sterol enrichment, thus explaining the apparent restriction to nLD formation we had observed (Figure 2C). High sterol levels in the membrane have never been reported as an elicitor of nLD formation. If our interpretation of the transcriptomic data is correct, increasing the presence of free sterols embedded in membranes should elicit nLD accumulation. We tested this last hypothesis by counting the percentage of cells in the population bearing nLD at the steady-state in strains that either cannot esterify and thus store sterols in cytoplasmic LD (*are1∆ are2∆*, simplified in the literature as *ste∆*) [28]; or that cannot hydrolyze sterols away from the membrane (*yeh2∆*) [29]. Importantly, in both mutants, the basal level of cells displaying nLD was increased (Figure 4E).

In conclusion, we propose that a profile of unsaturated phospholipids and high sterol levels at the nuclear membrane is supportive for the generation of nLD and thus responsive to the stimulation elicited by the attachment of stalled replication forks.

## 4. Discussion

Our understanding of genome stability preservation is shaped by our vast knowledge of the mechanism of DNA damage signaling, repair and replication. The importance of the spatial distribution of these phenomena has been progressively recognized, with a particular emphasis on the role of the nuclear membrane. First, the chromosomes occupy defined territories where the proximity to the nuclear membrane can define heterochromatin patterns [30,31]. As such, the increase in nuclear membrane extension, as when the nucleus adopts a multilobulated shape during neutrophil specification, creates novel heterochromatic territories, thus reshaping the epigenetic landscape with a functional meaning for immune cell performance [32]. Second, the relocalization of DNA molecules to the subdomains populated by nuclear pores anchors unrepairable broken ends [33], elicits recombination at stalled replication forks [18] and dictates mutation rates that can have adaptive consequences [34], among others. Last, the shape of the nuclear membrane has been matched to the degree of euploidy, evoking an additional, yet less understood link [35]. Thus, the crosstalk between chromatin and the nuclear membrane is central to the definition of genome homeostasis.

The nuclear membrane, mostly due to its lipid constituents, displays the potential to give rise to structures that invade the nucleoplasmic space, such as nucleoplasmic reticulum arms and nLD [5,21,36,37]. The generation of these structures has two immediate consequences: (1) they increase the surface of the membrane onto which DNA-related events could anchor or nucleate; (2) they occupy a space within the nucleoplasm, thus either forcing the nuclear volume to increase, or raising the intra-nuclear pressure. Both events will impact nuclear transactions [38]. Yet, the situations giving rise to nucleoplasmic reticulum intrusions and to nLD, and the impact they may have in genome homeostasis, are poorly defined.

To contribute to this characterization, in this work we have exploited *S. cerevisiae* as a model to evaluate the potential of different genotoxins to trigger nLD emission. We have found that only some genotoxins elicit a robust nLD formation in this model organism. Interestingly, these agents, namely HU, MMS and, more mildly, CPT (Figure 1), specifically challenge progressing replication forks. These observations suggest that nLD induction is not a feature of DNA damage per se, but may relate to a physical interaction between the DNA molecule, compromised during replication, and the membrane from which the nLD arise. In support, alternative means for creating replication stress by removing the proteins Exo1 or Sgs1 are also accompanied by nLD emission and, reciprocally, Exo1 overexpression suppresses replication stress-associated nLD formation (Figure 2). Of note, the mean number of nLD per cell remained close to 1 irrespective of the treatment (Appendix A), suggesting a potential focal concentration of challenged forks at a few INM locations. Which type of interaction could this be? The intrusion of nuclear membrane-derived structures, such as the nucleoplasmic reticulum, are induced by the pulling force of proteins simultaneously tethered to the membrane and to chromosomes, as is the case for condensins, the cohibin/CLIP complex or the ProMyelocytic Leukemia (PML) protein [5,20,21]. Since replication forks undergoing remodeling or repair are also reported to localize to the nuclear periphery [16,17,18,19], it could be that these contacts also pull the nuclear membrane inwards, thus setting physical parameters at the membrane apt for nLD nucleation and budding. This conforms with our previous report using different cultured human cells in which MMS also displayed this ability [36]. Yet, using these models, we failed to observe this same property for HU. In human cells, HU is known to lead to DNA breakage because of stalled replication fork processing [39], a situation that would immediately abrogate membrane pulling, thus explaining the different phenotype between yeast and human cells. Of note, bacterial genotoxins that trigger replication stress were recently shown to promote the emission towards the nucleoplasm of nucleoplasmic reticulum arms [40].

We also have explored the lipid membrane constituents that could be most supportive of nLD emission during replicative stress. We discard the contribution of PA (Figure 3), while we point to an increase in the unsaturation level of the membrane phospholipids as a possible elicitor of nLD emission (Figure 4). Further, we identify sterols, for the first time, as a novel contributor to nLD birth. Unsaturated fatty acids and sterols provide a loosely packed membrane that is more permissive to deformations. As such, pulling forces induced by attached DNA structures may have more chances to seed nLD birth events. This propensity to nLD emission from sterol-rich membranes matches a previous report in which lipidomic analysis of purified nLD revealed free sterols as major nLD constituents [41]. From a kinetic point of view, one could imagine that, in the first hour of treatment, MMS both stresses forks, which relocate to the INM and stimulate nLD birth, and also triggers the transcriptional downregulation program whose outcome is decreased sterol and lower unsaturation levels at the INM (Figure 4). Since the impact of this downregulation on the actual composition of the INM and its consequent limitation of nLD formation is not immediate, this would explain why nLD formation is more readily detected in the first 2 h of treatment than in the following ones (Figure 1D and Figure 2C,D).

Are nLD therefore harmful, somehow accelerating the degeneration of the genome? Or are they simply a by-product, for example of the transcriptional changes that ensue upon loss of genome integrity? Or could it be that, under certain conditions that challenge the DNA, nLD are beneficial? We recently observed that invasion of the nuclear space by voluminous globular bodies of cytoplasmic origin, which at the same time constrict chromatin and provide additional nuclear membrane surface, correlate with a faster activation of the DNA damage response [42]. In the same line, we observe that an increase in the pool of phospholipids in contact with the nucleoplasm, for example nLD and nucleoplasmic reticulum arms, ensures a stronger DNA damage response activation that, importantly, translates into improved genome integrity [43]. Together, the emerging picture is that phospholipid-coated structures in the nucleoplasm may serve as nucleating platforms that, through their interaction with chromatin [44], could launch DNA reactions. It has already been proposed that one of them is transcription modulation [4,45,46]. We now propose that stalled replication forks relocated to the nuclear periphery stimulate, through their attachment, the formation of nLD. Their occupancy of the nucleoplasmic space would, in turn, serve to nucleate factors involved in the DNA damage response activation (Figure 5), thus having an impact on genome homeostasis and the cell’s ability to proliferate.

## Figures and Tables

**Figure 1 cells-11-01390-f001:**
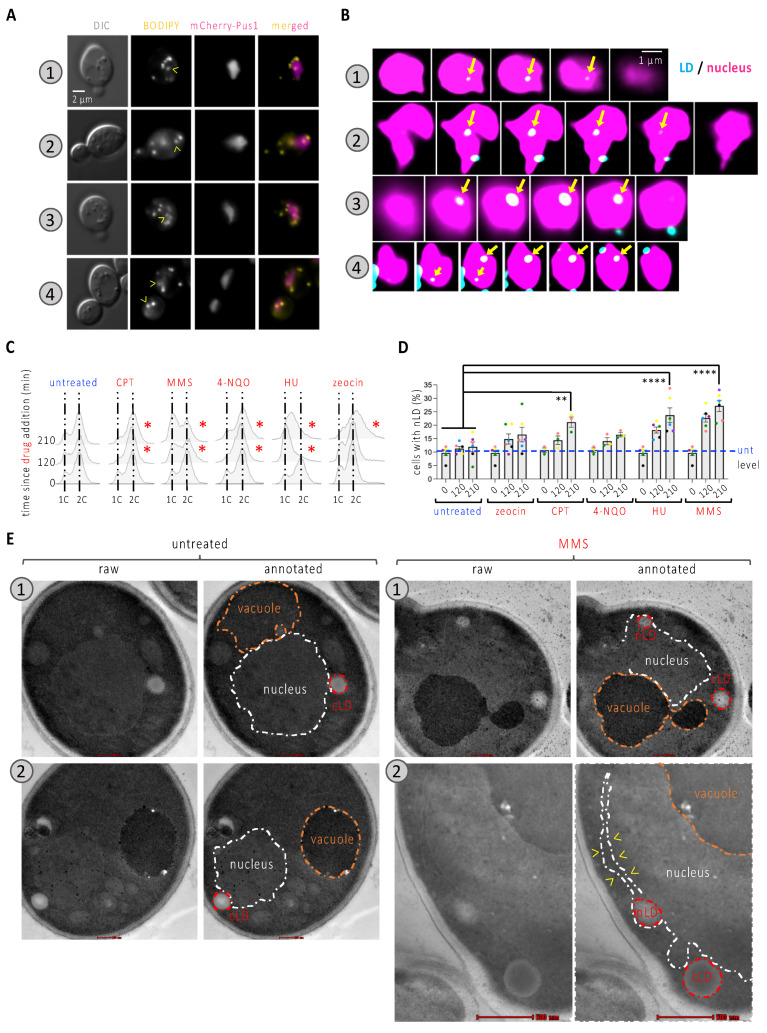
Characterization of nLD occurrence in response to genotoxins. (**A**) Cells expressing the mCherry-tagged Pus1 nucleoplasmic protein were also stained with BODIPY to dye LD. Arrowheads point at LD colocalizing with the nucleus. Numbers refer to 4 different examples. (**B**) Through-focus series were acquired every 200 nm across nuclei; images treated with Huygens software and consecutive slices are presented to illustrate the inclusion of the indicated LD fully within the nucleus (arrows). (**C**) Cytometry analysis of exponentially growing WT cells that were exposed to the indicated treatments for the indicated times. Delays as compared to the untreated condition are highlighted by a red asterisk. The used concentrations were 100 µM CPT; 0.1% MMS; 0.5 mg/L 4-NQO; 100 mM HU; 100 µg/mL zeocin. 1C and 2C refer to the DNA content. (**D**) The same cells from (**C**) were imaged as in (**A**) and the experimenter counted the percentage of cells displaying at least one LD in the nucleus (nLD). Each colored dot indicates one independently performed experiment. Dots of the same color allow for the verification of a trend and the reproducibility in response to a given treatment. The bar heights represent the mean of those experiments, and the error bars account for the Standard Error of the Mean (SEM). A multiple-comparison, paired one-way ANOVA was applied as indicated and the significance of the difference of the means is indicated by asterisks. **, *p* < 0.01; ****, *p* < 0.0001. At least 300 cells were counted per condition, time point and experiment. The level of the untreated samples (unt) is provided for comparison, indicated by a blue dashed line. (**E**) Samples treated with 0.1% MMS for 210 min and untreated samples were fixed and analyzed by electron microscopy. Two examples are shown per condition. Each image has been duplicated, one of which is annotated to highlight the structures of interest, such as the nucleus (white dashed line); the vacuole (orange dashed line); and the cytoplasmic and nuclear LD (red dashed line). The red bar size represents 200 nm except for example MMS.2, in which it represents 500 nm.

**Figure 2 cells-11-01390-f002:**
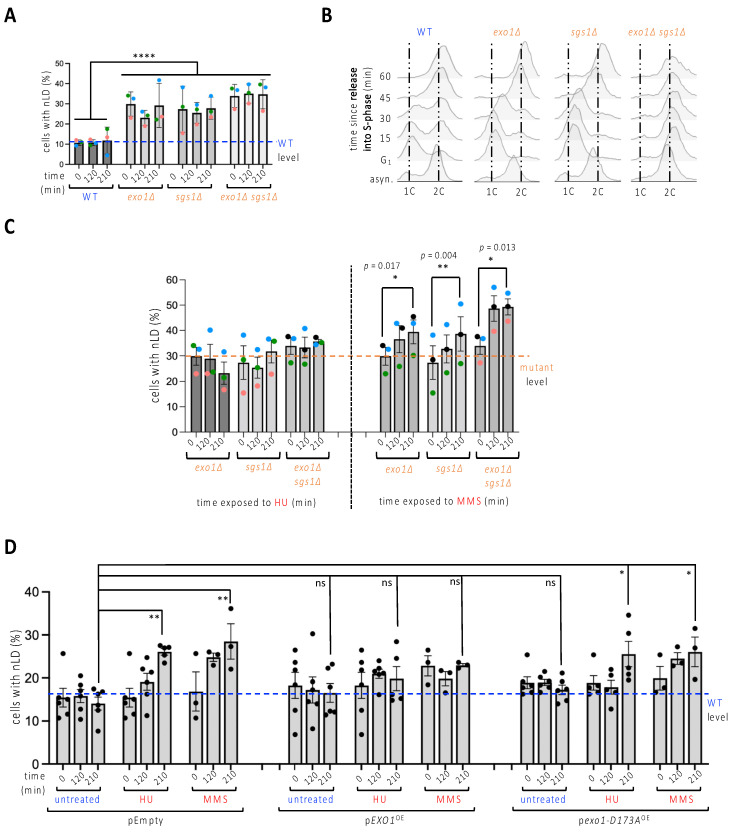
nLD occurrence in response to genetically induced replicative stress. (**A**) Exponentially growing cells of the indicated genotypes were imaged and the experimenter counted the percentage of cells displaying at least one LD in the nucleus (nLD). The measurement of the values at three different times in the absence of any treatment aims to assess whether time has any impact on the frequency of nLD. Each colored dot accounts for one independently performed experiment. Dots of the same color allow for the verification of a trend and the reproducibility in response to a given treatment. The bar heights represent the mean of those independent experiments, and the error bars account for the SEM. A multiple-comparison, one-way ANOVA was applied as indicated and the significance of the difference of the means is indicated by asterisks, where ****, *p* < 0.0001. At least 300 cells were counted per condition, time point and experiment. The level of the WT samples is provided for comparison, indicated by a blue dashed line. (**B**) Cytometry analysis of exponentially growing cells of the indicated genotypes that were collected either from asynchronous cultures (asyn.), or after a 2h30-long synchronization in G_1_ with alpha-factor, then allowed to progress into S-phase by removal of the alpha-factor block. 1C and 2C refer to the DNA content. Note that the double mutant *sgs1∆exo1∆* could not be synchronized in this timeframe, since all cells remained accumulated in G_2_ throughout the experiment. This may relate to a high basal accumulation of stressed replication forks and/or accumulated DNA damage as a consequence of having undergone replication for many generations in the combined absence of these key proteins. (**C**) Identical set-up as in (**A**) but cells were exposed either to 100 mM HU or to 0.1% MMS for the indicated times. In this case, a *t*-test analysis was performed using the three means of each independent experiment to compare the potential difference of the means for each strain between its untreated and its 210 min treatment conditions. Only significant differences are indicated by *p*-values. At least 300 cells were counted per condition, time point and experiment. The level of the mutant samples is provided for comparison, indicated by an orange dashed line. (**D**) Cells grown until the exponential phase in medium selective for the indicated plasmids (pEmpty, no gene; p*EXO1*^OE^, to overexpress the nuclease Exo1; p*exo1-D173A*^OE^, to overexpress a nuclease-dead Exo1 protein) were treated as stated for the indicated times, imaged as in Figure 1A, and scored to determine the frequency of nLD in the population. At least five independent experiments were performed, each represented by a black dot. The bar heights represent the mean of those experiments, and the error bars account for the SEM. A multiple-comparison one-way ANOVA was applied as indicated and the significance of the difference of the means is shown by asterisks, where *, *p* < 0.05; **, *p* < 0.01. At least 300 cells were counted per condition, time point and experiment. The level of the WT samples is provided for comparison, indicated by a blue dashed line.

**Figure 3 cells-11-01390-f003:**
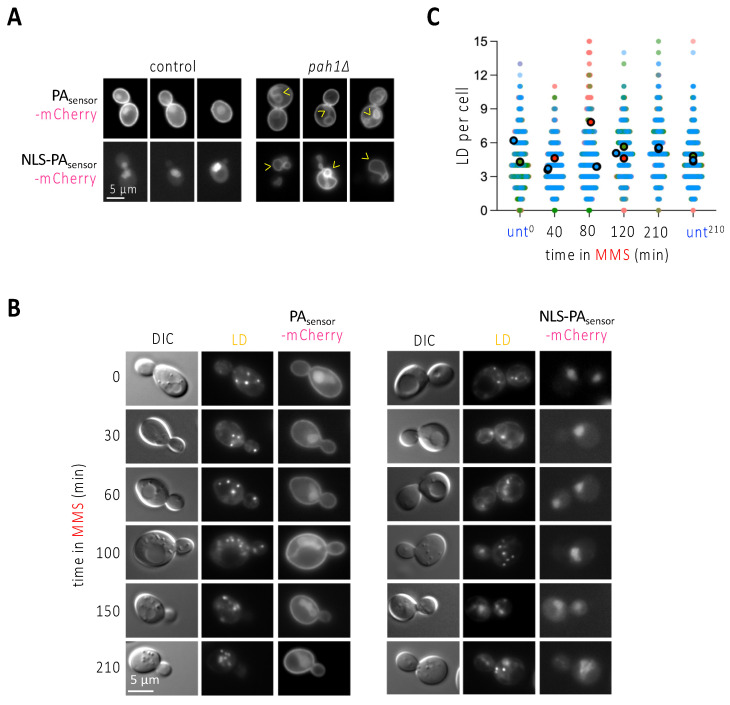
MMS treatment does not induce any apparent change in phosphatidic acid sub-cellular localization. (**A**) WT or cells lacking the phosphatidic acid (PA) phosphatase Pah1 were transformed with the indicated biosensors, aimed at detecting PA either all over the cell (PA_sensor_-mCherry), or specifically in the nucleus (NLS-PA_sensor_-mCherry). Three different examples of the patterns observed when imaging these cells are shown. Arrowheads point at deformed nuclear envelopes positive for the biosensor binding. (**B**) The exponentially growing WT cells described in (**A**) were exposed to 0.1% MMS for the indicated times and imaged to observe LD and the pattern yielded by the two PA biosensors. (**C**) WT cells treated with 0.1% MMS for the indicated times were stained with NileRed, and the total number of LD per cell scored. Three independent experiments were performed; each experiment is indicated by dots of a different color to allow reproducibility to be assessed [11]. The bigger, darker dots correspond to the mean of each of those independent experiments. At least 150 cells were counted per condition, time point and experiment.

**Figure 4 cells-11-01390-f004:**
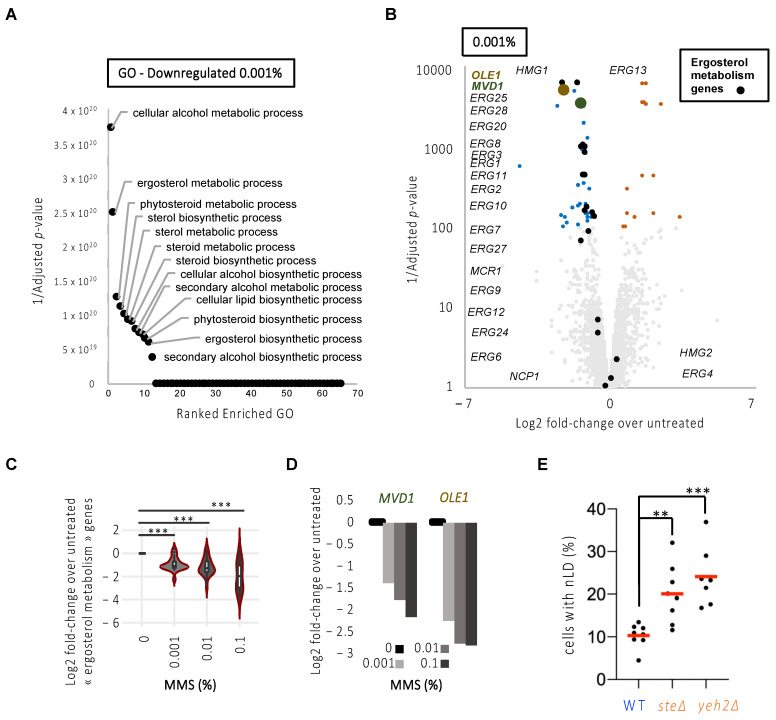
MMS treatment triggers a downregulation of fatty acid unsaturation and the sterol transcriptional network that is supportive of nLD formation. (**A**) Overrepresentation test of downregulated genes upon 1 h 0.001% MMS treatment. Enrichment is computed with Fisher’s test and corrected for multiple testing by false discovery rate. The 1/Adjusted *p*-values for all the enriched pathways are represented. The top pathways are annotated. See also Appendix A. (**B**) Volcano plots of all genes in response to 1 h 0.001% MMS. Fold-changes are computed from the average RNA signal of MMS-treated samples versus the average RNA signal of untreated ones. Adjusted *p*-values were computed with the Limma software [9] and corrected for multiple tests. Genes with significant changes according to the adjusted *p*-values are depicted in color, where upregulated ones are shown in orange and downregulated ones in blue. Genes annotated as related to ergosterol metabolism are annotated and depicted in black. *OLE1* and *MVD1* are represented in brown and green, respectively. See also Appendix A. (**C**) Violin plot for the average of the Log2 fold-change of the ergosterol-related genes for 1 h treatment with the three MMS doses under study. The asterisks indicate that *p* < 0.001 after applying a paired Mann–Whitney-test. (**D**) Bar plot for the average of the Log2 fold-change of *MVD1* and *OLE1* for 1 h treatment with the three MMS doses under study. (**E**) Exponentially growing cells of the indicated genotypes were imaged as in Figure 1A, and the experimenter counted the percentage of cells displaying at least one nLD. Each dot represents an independently performed experiment. The red bars represent the mean of those experiments. A multiple-comparison one-way ANOVA was applied as indicated and the significance of the difference of the means is stated by asterisks. **, *p* < 0.01; ***, *p* < 0.001. At least 300 cells were counted per strain and experiment.

**Figure 5 cells-11-01390-f005:**
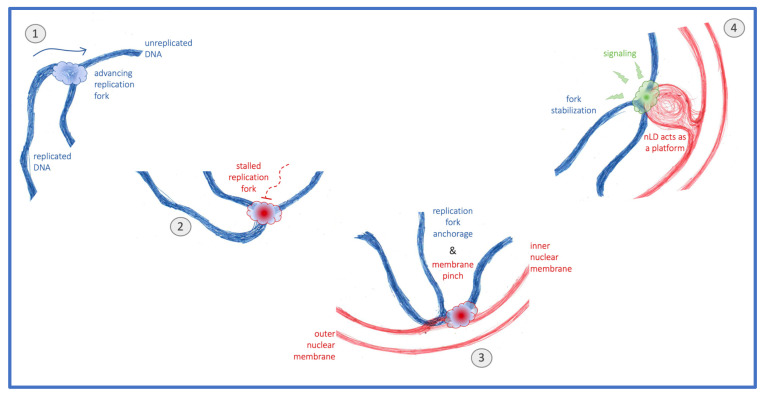
Model. The normal progression of replication forks (1) can be hampered by replicative stress. These stalled replication forks (2) relocate to the nuclear periphery, where they anchor, a process that can pinch the inner nuclear membrane (3). Whenever this membrane is rich in unsaturated phospholipids and sterols, fork attachment may elicit nuclear lipid droplet (nLD) formation, which subsequently act as platforms supporting stress signaling and downstream fork stabilization (4).

**Table 1 cells-11-01390-t001:** Strains used in this study.

Simplified Genotype	Full Genotype	Source
WT (W303)	*MAT* a, *ade2, his3, can1, leu2, trp1, ura3*, GAL+, psi+, *RAD5*	PP870, Philippe Pasero
*sgs1∆*	*MAT* a, *ade2, his3, can1, leu2, trp1, ura3, sgs1∆LEU2 mCherry-PUS1::URA3*	MM-113, Philippe Pasero
*sgs1∆ exo1∆*	*MAT* a, *ade2, his3, can1, leu2, trp1, ura3, sgs1∆LEU2 exo1∆ nat^R^ mCherry-PUS1::URA3*	MM-119, Philippe Pasero
*exo1∆*	*MAT* a, *ade2, his3, leu2, trp1, ura3*, *exo1∆nat^R^ mCherry-PUS1::URA3*	MM-110, Philippe Pasero
*ste∆*	*MAT* alpha, *ade2, his3, can1, leu2, trp1, ura3, are1∆HIS3, are2∆LEU2, mCherry-PUS1::URA3*	MM-55
*yeh2∆*	*MAT* a, *his3Δ1, leu2Δ0, met15Δ0, ura3Δ0, yeh2ΔkanMX6, mCherry-PUS1::URA3*	MM-60
WT (BY) mCherry-Pus1	*MAT* a, *his3Δ1, leu2Δ0, met15Δ0, ura3Δ0, mCherry-PUS1::URA3*	MM-98
WT (W303) mCherry-Pus1	*MAT* a, *ade2, his3, can1, leu2, trp1, ura3*, GAL+, psi+, *RAD5?, mCherry-PUS1::URA3*	MM-102

**Table 3 cells-11-01390-t003:** Differentially expressed top-30 hit genes in response to MMS (FC, fold-change).

Rank	Gene	Function	logFC_0.001	logFC_0.01	logFC_0.1	Average
1	** *HMG1* **	3 Hydroxy 3 MethylGlutaryl coenzyme a reductase	−2.2	−2.8	−3.0	−2.7
2	*OLE1*	OLEic acid requiring	−2.2	−2.8	−2.8	−2.6
3	*NDE1*	NADH Dehydrogenase, External	−2.1	−2.5	−2.5	−2.4
4	*COX7*	Cytochrome c OXidase	−1.5	−1.7	−3.4	−2.2
5	** *ERG28* **	ERGosterol biosynthesis	−1.2	−1.8	−3.5	−2.2
6	** *ERG3* **	ERGosterol biosynthesis	−1.1	−1.7	−3.7	−2.2
7	*IZH1*	Implicated in Zinc Homeostasis	−1.7	−2.3	−2.4	−2.1
8	*COX4*	Cytochrome c OXidase	−1.2	−1.5	−3.7	−2.1
9	*COX5A*	Cytochrome c OXidase	−1.5	−2.0	−2.6	−2.0
10	** *ERG25* **	ERGosterol biosynthesis	−1.2	−1.5	−3.3	−2.0
11	** *ERG2* **	ERGosterol biosynthesis	−1.2	−1.8	−3.0	−2.0
12	** *ERG13* **	ERGosterol biosynthesis	−1.5	−1.6	−2.7	−1.9
13	*COX12*	Cytochrome c OXidase	−1.4	−1.7	−2.6	−1.9
14	** *ERG11* **	ERGosterol biosynthesis	−1.1	−1.7	−2.8	−1.9
15	*HYP2*	HYPusine containing protein	−1.1	−1.7	−2.7	−1.8
16	*ACS2*	Acetyl CoA Synthetase	−1.3	−1.7	−2.5	−1.8
17	*MED1*	MEDiator complex	−1.7	−2.1	−1.6	−1.8
18	** *MVD1* **	MeValonate pyrophosphate Decarboxylase	−1.4	−1.8	−2.1	−1.8
19	** *ERG1* **	ERGosterol biosynthesis	−1.1	−1.4	−2.7	−1.7
20	*MTC7*	Maintenance of Telomere Capping	−1.1	−1.6	−2.4	−1.7
21	*SCW11*	Soluble Cell Wall protein	−1.1	−1.9	−2.0	−1.7
22	*CYT1*	CYTochrome c1	−1.2	−1.9	−1.7	−1.6
23	*QCR7*	ubiQuinol cytochrome C oxidoReductase	−1.0	−1.3	−2.0	−1.4
24	*QCR9*	ubiQuinol cytochrome C oxidoReductase	−1.1	−1.2	−2.0	−1.4
25	*NOP10*	NucleOlar Protein	−1.1	−1.2	−1.7	−1.3
26	** *ERG7* **	ERGosterol biosynthesis	−1.1	−1.2	−1.7	−1.3
27	** *ERG10* **	ERGosterol biosynthesis	−1.1	−1.2	−1.4	−1.2
28	*MSC7*	Meiotic Sister Chromatid recombination	−0.9	−1.2	−1.6	−1.2
29	** *ERG20* **	ERGosterol biosynthesis	−1.1	−1.3	−1.2	−1.2
30	** *ERG8* **	ERGosterol biosynthesis	−1.3	−1.4	−0.8	−1.2

## Data Availability

All the relevant data concerning this study are presented within this manuscript.

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
