# Peer review of "Nuclear Lipid Droplet Birth during Replicative Stress"

_cells, 2022, doi:10.3390/cells11091390_

Round 1

Reviewer 1 Report

In this article, the authors assess the formation of nuclear lipid droplets (nLD) as a result of genotoxin sensitivity. They use budding yeast as a model system and suggest a mechanism by which replication stress induced tethering of replication fork to nuclear membrane may prime the emission of nLD. They further show that unsaturated fatty acids and sterols contribute to the emission of nLD. Finally, they suggest the existence of a feedback mechanism by which MMS treatment further induces a transcriptional program resulting in the decrease of unsaturated fatty acids, thus saturating the formation of nLD. The findings here are interesting and provide some insight into the possible mechanism of nLD formation and suggests crosstalk between chromatin and nuclear membrane.

General Comments: The manuscript is well written, with clear logic and interpretation. The data in general are supportive to the major conclusions.

  • It would be informative if the authors could comment/discuss the significant increase in nLD formation after CPT treatment. Has this phenomenon been observed in other systems?
  • Authors suggest that nLD formation saturates at 50% after MMS treatment. But from what can be interpreted (as half of the figure is cut), it appears that it is seen in exo1∆/sgs1∆ mutant background. Do the authors see this saturation in Wild-Type (WT) cells? If yes, under what conditions?
  • The data-set used for the bioinformatics analysis for Figure4 is obtained after MMS treatment. How long after MMS treatment are the changes in transcriptional program observed? It would be nice if the authors could relate this with the nLD saturation observed in WT cells.

Specific Comments:

  • To understand the significance and reproducibility of the data, can the authors indicate the number of biological replicates used per genotype?
  • Parts of Figure 2C and 2D are cut in the version provided, making it difficult to review the conclusions for these figures.
  • Figure 2B: To conclude that the time taken to complete S phase is similar for WT and exo1∆/sgs1∆ mutant cells, please show a complete FACS profile with samples taken after regular intervals following arrest and release from G1.
  • Line 259-260: No increase observed after HU addition. Can the authors comment on why they suggest a mild increase is observed after HU treatment?
  • Line 262: figure label is incorrect

Reviewer 2 Report

This is an interesting paper linking nLD formation with DNA replication stress and nuclear envelope lipid alteration.However, little data on DNA-replication fork anchorage with nuclear envelope as well as nuclear envelope pinching due to replication stress, possibly related to nLD formation, has been shown.

Major Comments 1) Please add information about the size as a criterion how you recognize the bodipy/nilered-positive structure as an nLD: Please state in the text how large of the bodipy/nilered-positive structure you perceived as an nLD. In other words, what is limitation of the LD detection? 2) Was there a difference in the number of nLDs due to the difference in DNA stress? If so, in Fig. 1D, was a difference in the number of nLDs considered?

Miner comments 1) Fig. 2 was not readable because half of it was not printed. 2) line 41-42: please check the sentence. 3) Please describe how long the MMS-exposure was performed when you obtained the transcriptome analysis in Fig. 4.

The referee guess that it was 120 min or 210 min as described in Fig. 1 and Fig. 2. It may have been written somewhere in the text and I couldn't find it. 
